# MintNet: Building Invertible Neural Networks with Masked Convolutions

**Yang Song**[*]
Stanford University
yangsong@cs.stanford.edu

**Chenlin Meng**[*]
Stanford University
chenlin@cs.stanford.edu

**Stefano Ermon**
Stanford University
ermon@cs.stanford.edu

## Abstract

We propose a new way of constructing invertible neural networks by combining simple building blocks with a novel set of composition rules. This leads to a rich set of invertible architectures, including those similar to ResNets. Inversion is achieved with a locally convergent iterative procedure that is parallelizable and very fast in practice. Additionally, the determinant of the Jacobian can be computed analytically and efficiently, enabling their generative use as flow models. To demonstrate their flexibility, we show that our invertible neural networks are competitive with ResNets on MNIST and CIFAR-10 classification. When trained as generative models, our invertible networks achieve competitive likelihoods on MNIST, CIFAR-10 and ImageNet $32\times32$, with bits per dimension of 0.98, 3.32 and 4.06 respectively.

## 1  Introduction

Invertible neural networks have many applications in machine learning. They have been employed to investigate representations of deep classifiers [15], understand the cause of adversarial examples [14], learn transition operators for MCMC [28, 18], create generative models that are directly trainable by maximum likelihood [6, 5, 24, 16, 9, 1], and perform approximate inference [27, 17].

Many applications of invertible neural networks require that both inverting the network and computing the Jacobian determinant be efficient. While typical neural networks are not invertible, achieving these properties often imposes restrictive constraints to the architecture. For example, planar flows [27] and Sylvester flow [2] constrain the number of hidden units to be smaller than the input dimension. NICE [5] and Real NVP [6] rely on dimension partitioning heuristics and specific architectures such as coupling layers, which could make training more difficult [1]. Methods like FFJORD [9], i-ResNets [1] have fewer architectural constraints. However, their Jacobian determinants have to be approximated, which is problematic if repeatedly performed at training time as in flow models.

In this paper, we propose a new method of constructing invertible neural networks which are flexible, efficient to invert, and whose Jacobian can be computed *exactly* and efficiently. We use triangular matrices as our basic module. Then, we provide a set of composition rules to recursively build more complex non-linear modules from the basic module, and show that the composed modules are invertible as long as their Jacobians are non-singular. As in previous work [6, 24], the Jacobians of our modules are triangular, allowing efficient determinant computation. The inverse of these modules can be obtained by an efficiently parallelizable fixed-point iteration method, making the cost of inversion comparable to that of an i-ResNet [1] block.

Using our composition rules and masked convolutions as the basic triangular building block, we construct a rich set of invertible modules to form a deep invertible neural network. The architecture of our proposed invertible network closely follows that of ResNet [10]—the state-of-the-art architecture

---

[*]Equal contribution.

of discriminative learning. We call our model **M**asked **In**vertible **Net**work (MintNet). To demonstrate the capacity of MintNets, we first test them on image classification. We found that a MintNet classifier achieves 99.6% accuracy on MNIST, matching the performance of a ResNet with a similar architecture. On CIFAR-10, it achieves 91.2% accuracy, comparable to the 92.6% accuracy of ResNet. When using MintNets as generative models, they achieve the new state-of-the-art results of bits per dimension (bpd) on uniformly dequantized images. Specifically, MintNet achieves bpd values of 0.98, 3.32, and 4.06 on MNIST, CIFAR-10 and ImageNet $32\times32$, while former best published results are 0.99 (FFJORD [9]), 3.35 (Glow [16]) and 4.09 (Glow) respectively. Moreover, MintNet uses fewer parameters and less computational resources. Our MNIST model uses 30% fewer parameters than FFJORD [9]. For CIFAR-10 and ImageNet $32\times32$, MintNet uses 60% and 74% fewer parameters than the corresponding Glow [16] models. When training on dataset such as CIFAR-10, MintNet required 2 GPUs for approximately 5 days, while FFJORD [9] used 6 GPUs for approximately 5 days, and Glow [16] used 8 GPUs for approximately 7 days.

## 2 Background

Consider a neural network $f : \mathbb{R}^D \to \mathbb{R}^L$ that maps a data point $\mathbf{x} \in \mathbb{R}^D$ to a latent representation $\mathbf{z} \in \mathbb{R}^L$. When for every $\mathbf{z} \in \mathbb{R}^L$ there exists a unique $\mathbf{x} \in \mathbb{R}^D$ such that $f(\mathbf{x}) = \mathbf{z}$, we call $f$ an invertible neural network. There are several basic properties of invertible networks. First, when $f(\mathbf{x})$ is continuous, a necessary condition for $f$ to be invertible is $D = L$. Second, if $f_1 : \mathbb{R}^D \to \mathbb{R}^D$ and $f_2 : \mathbb{R}^D \to \mathbb{R}^D$ are both invertible, $f = f_2 \circ f_1$ will also be invertible. In this work, we mainly consider applications of invertible neural networks to classification and generative modeling.

### 2.1 Classification with invertible neural networks

Neural networks for classification are usually not invertible because the number of classes $L$ is usually different from the input dimension $D$. Therefore, when discussing invertible neural networks for classification, we separate the classifier into two parts $f = f_2 \circ f_1$: feature extraction $\mathbf{z} = f_1(\mathbf{x})$ and classification $\mathbf{y} = f_2(\mathbf{z})$, where $f_2$ is usually the softmax function. We say the classifier is invertible when $f_1$ is invertible. Invertible classifiers are arguably more interpretable, because a prediction can be traced down by inverting latent representations [15, 14].

### 2.2 Generative modeling with invertible neural networks

An invertible network $f : \mathbf{x} \in \mathbb{R}^D \mapsto \mathbf{z} \in \mathbb{R}^D$ can be used to warp a complex probability density $p(\mathbf{x})$ to a simple base distribution $\pi(\mathbf{z})$ (*e.g.*, a multivariate standard Gaussian) [5, 6]. Under the condition that both $f$ and $f^{-1}$ are differentiable, the densities of $p(\mathbf{x})$ and $\pi(\mathbf{z})$ are related by the following change of variable formula

$$\log p(\mathbf{x}) = \log \pi(\mathbf{z}) + \log |\det(J_f(\mathbf{x}))|, \tag{1}$$

where $J_f(\mathbf{x})$ denotes the Jacobian of $f(\mathbf{x})$ and we require $J_f(\mathbf{x})$ to be non-singular so that $\log |\det(J_f(\mathbf{x}))|$ is well-defined. Using this formula, $p(\mathbf{x})$ can be easily computed if the Jacobian determinant $\det(J_f(\mathbf{x}))$ is cheaply computable and $\pi(\mathbf{z})$ is known.

Therefore, an invertible neural network $f_{\boldsymbol{\theta}}(\mathbf{x})$ implicitly defines a normalized density model $p_{\boldsymbol{\theta}}(\mathbf{x})$, which can be directly trained by maximum likelihood. The invertibility of $f_{\boldsymbol{\theta}}$ is critical to fast sample generation. Specifically, in order to generate a sample $\mathbf{x}$ from $p_{\boldsymbol{\theta}}(\mathbf{x})$, we can first draw $\mathbf{z} \sim \pi(\mathbf{z})$, and warp it back through the inverse of $f_{\boldsymbol{\theta}}$ to obtain $\mathbf{x} = f_{\boldsymbol{\theta}}^{-1}(\mathbf{z})$.

Note that multiple invertible models $f_1, f_2, \cdots, f_K$ can be stacked together to form a deeper invertible model $f = f_K \circ \cdots \circ f_2 \circ f_1$, without much impact on the inverse and determinant computation. This is because we can sequentially invert each component, *i.e.*, $f^{-1} = f_1^{-1} \circ f_2^{-1} \circ \cdots \circ f_K^{-1}$, and the total Jacobian determinant equals the product of each individual Jacobian determinant, *i.e.*, $|\det(J_f)| = |\det(J_{f_1})||\det(J_{f_2})| \cdots |\det(J_{f_K})|$.

## 3 Building invertible modules compositionally

In this section, we discuss how simple blocks like masked convolutions can be composed to build invertible modules that allow efficient, parallelizable inversion and determinant computation. To this

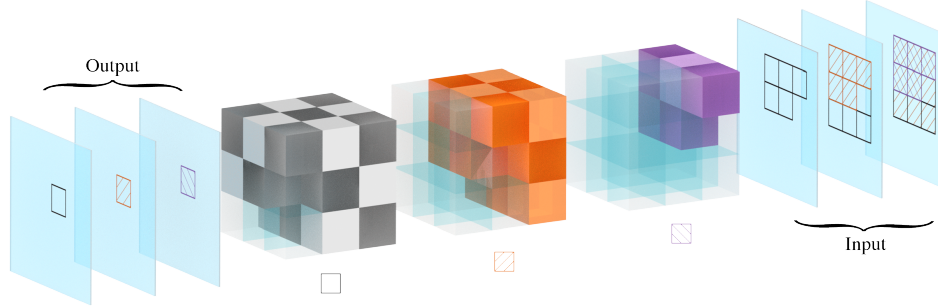

Figure 1: Illustration of a masked convolution with 3 filters and kernel size $3 \times 3$. Solid checkerboard cubes inside each filter represent unmasked weights, while the transparent blue blocks represent the weights that have been masked out. The receptive field of each filter on the input feature maps is indicated by regions shaded with the pattern (the colored square) below the corresponding filter.

end, we first introduce the basic building block of our models. Then, we propose a set of composition rules to recursively build up complex non-linear modules with triangular Jacobians. Next, we prove that these composed modules are invertible as long as their Jacobians are non-singular. Finally, we discuss how these modules can be inverted efficiently using numerical methods.

## 3.1 The basic module

We start from considering linear transformations $f(\mathbf{x}) = \mathbf{W}\mathbf{x} + \mathbf{b}$, with $\mathbf{W} \in \mathbb{R}^{D \times D}$, and $\mathbf{b} \in \mathbb{R}^D$. For a general $\mathbf{W}$, computing its Jacobian determinant requires $O(D^3)$ operations. We therefore choose $\mathbf{W}$ to be a triangular matrix. In this case, the Jacobian determinant $\det(J_f(\mathbf{x})) = \det(\mathbf{W})$ is the product of all diagonal entries of $\mathbf{W}$, and the computational complexity is reduced to $O(D)$. The linear function $f(\mathbf{x}) = \mathbf{W}\mathbf{x} + \mathbf{b}$ with $\mathbf{W}$ being triangular is our *basic module*.

**Masked convolutions.** Convolution is a special type of linear transformation that is very effective for image data. The triangular structure of the basic module can be achieved using *masked* convolutions (*e.g.*, causal convolutions in PixelCNN [22]). We provide the formula of our masks in Appendix B and an illustration of a $3 \times 3$ masked convolution with 3 filters in Fig. 1. Intuitively, the causal structure of the filters (ordering of the pixels) enforces a triangular structure.

## 3.2 The calculus of building invertible modules

Complex non-linear invertible functions can be constructed from our basic modules in two steps. First, we follow several composition rules so that the composed module has a triangular Jacobian. Next, we impose appropriate constraints so that the module is invertible. To simplify the discussion, we only consider modules with lower triangular Jacobians here, and we note that it is straightforward to extend the analysis to modules with upper triangular Jacobians.

The following proposition summarizes several rules to compositionally build new modules with triangular Jacobians using existing ones.

**Proposition 1.** *Define $\mathcal{F}$ as the set of all continuously differentiable functions whose Jacobian is lower triangular. Then $\mathcal{F}$ contains the basic module in Section 3.1, and is closed under the following composition rules.*

- ***Rule of addition**. $f_1 \in \mathcal{F} \wedge f_2 \in \mathcal{F} \Rightarrow \lambda f_1 + \mu f_2 \in \mathcal{F}$, where $\lambda, \mu \in \mathbb{R}$.*

- ***Rule of composition**. $f_1 \in \mathcal{F} \wedge f_2 \in \mathcal{F} \Rightarrow f_2 \circ f_1 \in \mathcal{F}$. A special case is $f \in \mathcal{F} \Rightarrow h \circ f \in \mathcal{F}$, where $h(\cdot)$ is a continuously differentiable non-linear activation function that is applied element-wise.*

The proof of this proposition is straightforward and deferred to Appendix A. By repetitively applying the rules in Proposition 1, our basic linear module can be composed to construct complex non-linear modules having continuous and triangular Jacobians. Note that besides our linear basic modules,

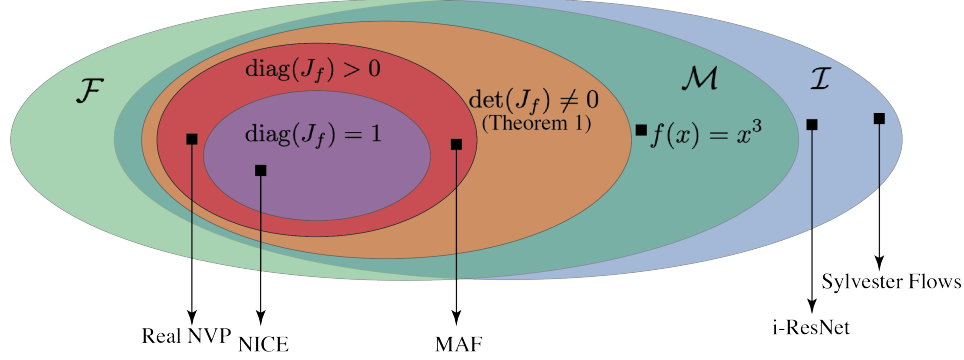

Figure 2: Venn Diagram relationships between invertible functions ($\mathcal{I}$), the function sets of $\mathcal{F}$ and $\mathcal{M}$, functions that meet the conditions of Theorem 1 ($\det(J_f) \neq 0$), functions whose Jacobian is triangular and Jacobian diagonals are strictly positive ($\mathrm{diag}(J_f) > 0$), functions whose Jacobian is triangular and Jacobian diagonals are all 1s ($\mathrm{diag}(J_f) = 1$).

other functions with triangular and continuous Jacobians can also be made more expressive using the composition rules. For example, the layers of dimension partitioning models (*e.g.*, NICE [5], Real NVP [6], Glow [16]) and autoregressive flows (*e.g.*, MAF [24]) all have continuous and triangular Jacobians and therefore belong to $\mathcal{F}$. Note that the rule of addition in Proposition 1 preserves triangular Jacobians but not invertibility. Therefore, we need additional constraints if we want the composed functions to be invertible.

Next, we state the condition for $f \in \mathcal{F}$ to be invertible, and denote the invertible subset of $\mathcal{F}$ as $\mathcal{M}$.

**Theorem 1.** *If $f \in \mathcal{F}$ and $J_f(\mathbf{x})$ is non-singular for all $\mathbf{x}$ in the domain, then $f$ is invertible.*

*Proof.* A proof can be found in Appendix A. □

The non-singularity of $J_f(\mathbf{x})$ constraint in Theorem 1 is natural in the context of generative modeling. This is because in order for Eq. (1) to make sense, $\log|\det(J_f)|$ has to be well-defined, which requires $J_f(\mathbf{x})$ to be non-singular.

In many cases, Theorem 1 can be easily used to check and enforce the invertibility of $f \in \mathcal{F}$. For example, the layers of autoregressive flow models and dimension partitioning models can all be viewed as elements of $\mathcal{F}$ because they are continuously differentiable and have triangular Jacobians. Since the diagonal entries of their Jacobians are always strictly positive and hence non-singular, we can immediately conclude that they are invertible with Theorem 1, thus generalizing their model-specific proofs of invertibility.

In Fig. 2, we provide a Venn Diagram to illustrate the set of functions that satisfy the condition of Theorem 1. As depicted by the orange set labeled by $\det(J_f) \neq 0$, Theorem 1 captures a subset of $\mathcal{M}$ where the Jacobians of functions are non-singular so that the change of variable formula is usable. Note the condition in Theorem 1 is sufficient but not necessary. For example, $f(x) = x^3 \in \mathcal{M}$ is invertible, but $J_f(x=0) = 3x^2|_{x=0} = 0$ is singular. Many previous invertible models with special architectures, such as NICE, Real NVP, and MAF, can be viewed as elements belonging to subsets of $\det(J_f) \neq 0$.

### 3.3 Efficient inversion of the invertible modules

In this section, we show that when the conditions in Theorem 1 hold, not only do we know that $f$ is invertible ($f \in \mathcal{M}$), but also we have a fixed-point iteration method to invert $f$ with strong theoretical guarantees and good performance in practice.

The pseudo-code of our proposed inversion algorithm is described in Algorithm 1. Theoretically, we can prove that this method is locally convergent—as long as the initial value is close to the true value, the method is guaranteed to find the correct inverse. We formally summarize this result in Theorem 2.

**Theorem 2.** *The iterative method of Algorithm 1 is locally convergent whenever $0 < \alpha < 2$.*

**Algorithm 1** Fixed-point iteration method for computing $f^{-1}(\mathbf{z})$.

---

**Require:** $T, \alpha$          ▷ $T$ is the number of iterations; $0 < \alpha < 2$ is the step size.
1: Initialize $\mathbf{x}_0$
2: **for** $t \leftarrow 1$ to $T$ **do**
3:     Compute $f(\mathbf{x}_{t-1})$
4:     Compute $\mathrm{diag}(J_f(\mathbf{x}_{t-1}))$
5:     $\mathbf{x}_t \leftarrow \mathbf{x}_{t-1} - \alpha \, \mathrm{diag}(J_f(\mathbf{x}_{t-1}))^{-1}(f(\mathbf{x}_{t-1}) - \mathbf{z})$
6: **end for**
   return $\mathbf{x}_T$

---

*Proof.* We provide a more rigorous proof in Appendix A.       □

In practice, the method is also easily parallelizable on GPUs, making the cost of inverting $f \in \mathcal{M}$ similar to that of an i-ResNet [1] layer. Within each iteration, the computation is mostly matrix operations that can be vectorized and run efficiently in parallel. Therefore, the time cost will be roughly proportional to the number of iterations, *i.e.*, $O(T)$. As will be shown in our experiments, Algorithm 1 converges fast and usually the error quickly becomes negligible when $T \ll D$. This is in stark contrast to existing methods of inverting autoregressive flow models such as MAF [24], where $D$ univariate equations need to be solved sequentially, requiring at least $O(D)$ iterations. There are also other approaches for inverting $f$. For example, the bisection method is guaranteed to converge globally, but its computational cost is $O(D)$, and is usually much more expensive than Algorithm 1. Note that as discussed earlier, autoregressive flow models can also be viewed as special cases of our framework. Therefore, Algorithm 1 is also applicable to inverting autoregressive flow models and could potentially result in large improvements of sampling speed.

## 4 Masked Invertible Networks

We show that techniques developed in Section 3 can be used to build our Masked Invertible Network (MintNet). First, we discuss how we compose several masked convolutions to form the Masked Invertible Layer (Mint layer). Next, we stack multiple Mint layers to form a deep neural network, *i.e.*, the MintNet. Finally, we compare MintNets with several existing invertible architectures.

### 4.1 Building the Masked Invertible Layer

We construct an invertible module in $\mathcal{M}$ that serves as the basic layer of our MintNet. This invertible module, named Mint layer, is defined as

$$\mathfrak{L}(\mathbf{x}) = \mathbf{t} \odot \mathbf{x} + \sum_{i=1}^{K} \mathbf{W}_i^3 h\left( \sum_{j=1}^{K} \mathbf{W}_{ij}^2 h(\mathbf{W}_j^1 \mathbf{x} + \mathbf{b}_j^1) + \mathbf{b}_{ij}^2 \right) + \mathbf{b}_i^3, \tag{2}$$

where $\odot$ denotes the elementwise multiplication, $\{\mathbf{W}_i^1\}|_{i=1}^{K}$, $\{\mathbf{W}_{ij}^2\}|_{1 \le i,j \le K}$, and $\{\mathbf{W}_i^3\}|_{i=1}^{K}$ are all lower triangular matrices with additional constraints to be specified later, and $\mathbf{t} > \mathbf{0}$. Additionally, Mint layers use a monotonic activation function $h$, so that $h' \ge 0$. Common choices of $h$ include ELU [4], tanh and sigmoid. Note that every individual weight matrix has the same size, and the 3 groups of weights $\{\mathbf{W}_i^1\}|_{i=1}^{K}$, $\{\mathbf{W}_{ij}^2\}|_{1 \le i,j \le K}$ and $\{\mathbf{W}_i^3\}|_{i=1}^{K}$ can be implemented with 3 masked convolutions (see Appendix B). We design the form of $\mathfrak{L}(\mathbf{x})$ so that it resembles a ResNet / i-ResNet block that also has 3 convolutions with $K \times C$ filters, with $C$ being the number of channels of $\mathbf{x}$. When using Algorithm 1 to invert Mint layers, we initialize $\mathbf{x}_0 = \mathbf{z} \odot \frac{1}{\mathbf{t}}$.

From Proposition 1 in Section 3.2, we can easily conclude that $\mathfrak{L} \in \mathcal{F}$. Now, we consider additional constraints on the weights so that $\mathfrak{L} \in \mathcal{M}$, *i.e.*, it is invertible. Note that the analytic form of its Jacobian is

$$J_{\mathfrak{L}}(\mathbf{x}) = \sum_{i=1}^{K} \mathbf{W}_i^3 \mathbf{A}_i \sum_{j=1}^{K} \mathbf{W}_{ij}^2 \mathbf{B}_j \mathbf{W}_j^1 + \mathbf{t}, \tag{3}$$

with $\mathbf{A}_i = \mathrm{diag}(h'\left(\sum_{j=1}^{K} \mathbf{W}_{ij}^2 h(\mathbf{W}_j^1 \mathbf{x} + \mathbf{b}_j^1) + \mathbf{b}_{ij}^2\right)) \geq \mathbf{0}$, $\mathbf{B}_j = \mathrm{diag}(h'(\mathbf{W}_j^1 \mathbf{x} + \mathbf{b}_j^1)) \geq \mathbf{0}$, and $\mathbf{t} > \mathbf{0}$. Therefore, once we impose the following constraint

$$\mathrm{diag}(\mathbf{W}_i^3)\,\mathrm{diag}(\mathbf{W}_{ij}^2)\,\mathrm{diag}(\mathbf{W}_j^1) \geq \mathbf{0}, \forall 1 \leq i, j \leq K, \tag{4}$$

we have $\mathrm{diag}(J_{\mathfrak{L}}(\mathbf{x})) > \mathbf{0}$, which satisfies the condition of Theorem 1 and as a consequence we know $\mathfrak{L} \in \mathcal{M}$. In practice, the constraint Eq. (4) can be easily implemented. For all $1 \leq i, j \leq K$, we impose no constraint on $\mathbf{W}_i^3$ and $\mathbf{W}_j^1$, but replace $\mathbf{W}_{ij}^2$ with $\mathbf{V}_{ij}^2 = \mathbf{W}_{ij}^2 \, \mathrm{sign}(\mathrm{diag}(\mathbf{W}_{ij}^2))\, \mathrm{sign}(\mathrm{diag}(\mathbf{W}_i^3 \mathbf{W}_j^1))$. Note that $\mathrm{diag}(\mathbf{V}_{ij}^2)$ has the same signs as $\mathrm{diag}(\mathbf{W}_i^3)\,\mathrm{diag}(\mathbf{W}_j^1)$ and therefore $\mathrm{diag}(\mathbf{W}_i^3)\,\mathrm{diag}(\mathbf{V}_{ij}^2)\,\mathrm{diag}(\mathbf{W}_j^1) \geq \mathbf{0}$. Moreover, $\mathbf{V}_{ij}^2$ is almost everywhere differentiable w.r.t. $\mathbf{W}_{ij}^2$, which allows gradients to backprop through.

## 4.2 Constructing the Masked Invertible Network

In this section, we introduce design choices that help stack multiple Mint layers together to form an expressive invertible neural network, namely the MintNet. The full MintNet is constructed by stacking the following paired Mint layers and squeezing layers.

**Paired Mint layers.** As discussed above, our Mint layer $\mathfrak{L}(\mathbf{x})$ always has a triangular Jacobian. To maximize the expressive power of our invertible neural network, it is undesirable to constrain the Jacobian of the network to be triangular since this limits capacity and will cause blind spots in the receptive field of masked convolutions. We thus always pair two Mint layers together—one with a lower triangular Jacobian and the other with an upper triangular Jacobian, so that the Jacobian of the paired layers is not triangular, and blind spots can be eliminated.

**Squeezing layers.** Subsampling is important for enlarging the receptive field of convolutions. However, common subsampling operations such as pooling and strided convolutions are usually not invertible. Following [6] and [1], we use a "squeezing" operation to reshape the feature maps so that they have smaller resolution but more channels. After a squeezing operation, the height and width will decrease by a factor of $k$, but the number of channels will increase by a factor of $k^2$. This procedure is invertible and the Jacobian is an identity matrix. Throughout the paper, we use $k = 2$.

## 4.3 Comparison to other approaches

In what follows we compare MintNets to several existing methods for developing invertible architectures. We will focus on architectures with a tractable Jacobian determinant. However, we note that there are models (*cf.*, [7, 21, 8]) that allow fast inverse computation but do not have tractable Jacobian determinants. Following [1], we also provide some comparison in Tab. 5 (see Appendix E).

### 4.3.1 Models based on identities of determinants

Some identities can be used to speed up the computation of determinants if the Jacobians have special structures. For example, in Sylvester flow [2], the invertible transformation has the form $f(\mathbf{x}) \triangleq \mathbf{x} + \mathbf{A}h(\mathbf{B}\mathbf{x} + \mathbf{b})$, where $h(\cdot)$ is a nonlinear activation function, $\mathbf{A} \in \mathbb{R}^{D \times M}$, $\mathbf{B} \in \mathbb{R}^{M \times D}$, $\mathbf{b} \in \mathbb{R}^M$ and $M \leq D$. By Sylvester's determinant identity, $\det(J_f(\mathbf{x}))$ can be computed in $O(M^3)$, which is much less than $O(D^3)$ if $M \ll D$. However, the requirement that $M$ is small becomes a bottleneck of the architecture and limits its expressive power. Similarly, Planar flow [27] uses the matrix determinant lemma, but has an even narrower bottleneck.

The form of $\mathfrak{L}(\mathbf{x})$ bears some resemblance to Sylvester flow. However, we improve the capacity of Sylvester flow in two ways. First, we add one extra non-linear convolutional layer. Second, we avoid the bottleneck that limits the maximum dimension of latent representations in Sylvester flow.

### 4.3.2 Models based on dimension partitioning

NICE [5], Real NVP [6], and Glow [16] all depend on an affine coupling layer. Given $d < D$, $\mathbf{x}$ is first partitioned into two parts $\mathbf{x} = [\mathbf{x}_{1:d}; \mathbf{x}_{d+1:D}]$. The coupling layer is an invertible transformation, defined as $f : \mathbf{x} \mapsto \mathbf{z}$, $\mathbf{z}_{1:d} = \mathbf{x}_{1:d}$, $\mathbf{z}_{d+1:D} = \mathbf{x}_{d+1:D} \odot \exp(s(\mathbf{x}_{1:d})) + t(\mathbf{x}_{1:d})$, where $s(\cdot)$ and $t(\cdot)$ are two arbitrary functions. However, the partitioning of $\mathbf{x}$ relies on heuristics, and the performance is sensitive to this choice (*cf.*, [16, 1]). In addition, the Jacobian of $f$ is a triangular

matrix with diagonal $[\mathbf{1}_d; \exp(s(\mathbf{x}_{1:d}))]$. In contrast, the Jacobian of MintNets has more flexible diagonals—without being partially restricted to 1's.

### 4.3.3 Models based on autoregressive transformations

By leveraging autoregressive transformations, the Jacobian can be made triangular. For example, MAF [24] defines the invertible tranformation as $f : \mathbf{x} \mapsto \mathbf{z}$, $\quad z_i = \mu(\mathbf{x}_{1:i-1}) + \sigma(\mathbf{x}_{1:i-1}) x_i$, where $\mu(\cdot) \in \mathbb{R}$ and $\sigma(\cdot) \in \mathbb{R}^+$. Note that $f^{-1}(\mathbf{z})$ can be obtained by sequentially solving $x_i$ based on previous solutions $\mathbf{x}_{1:i-1}$. Therefore, a naïve approach requires $\Omega(D)$ computations for inverting autoregressive models. Moreover, the architecture of $f$ is only an affine combination of autoregressive functions with $\mathbf{x}$. In contrast, MintNets are inverted with faster fixed-point iteration methods, and the architecture of MintNets is arguably more flexible.

### 4.3.4 Free-form invertible models

Some work proposes invertible transformations whose Jacobians are not limited by special structures. For example, FFJORD [9] uses a continuous version of change of variables formula [3] where the determinant is replaced by trace. Unlike MintNets, FFJORD needs an ODE solver to compute its value and inverse, and uses a stochastic estimator to approximate the trace. Another work is i-ResNet [1] which constrains the Lipschitz-ness of ResNet layers to make it invertible. Both i-ResNet and MintNet use ResNet blocks with 3 convolutions. The inverse of i-ResNet can be obtained efficiently by a parallelizable fixed-point iteration method, which has comparable computational cost as our Algorithm 1. However, unlike MintNets whose Jacobian determinants are exact, the log-determinant of Jacobian of an i-ResNet must be approximated by truncating a power series and estimating each term with stochastic estimators.

### 4.3.5 Other models using masked convolutions

Emerging convolutions [13] and MaCow [20] improve the Glow architecture by replacing $1 \times 1$ convolutions in the original Glow model with masked convolutions similar to those employed in MintNets. Emerging convolutions and MaCow are both inverted using forward/back substitutions designed for inverting triangular matrices, which requires the same number of iterations as the input dimension. In stark contrast, MintNets use a fixed-point iteration method (Algorithm 1) for inversion, which is similar to i-ResNet and requires substantially fewer iterations than the input dimension. For example, our method of inversion takes 120 iterations to converge on CIFAR-10, while inverting emerging convolutions will need 3072 iterations. In other words, our inversion can be 25 times faster on powerful GPUs. Additionally, the architecture of MintNet is very different. The architectures of [13] and [20] are both built upon Glow. In contrast, MintNet is a ResNet architecture where normal convolutions are replaced by causal convolutions.

## 5 Experiments

In this section, we evaluate our MintNet architectures on both image classification and density estimation. We focus on three common image datasets, namely MNIST, CIFAR-10 and ImageNet $32 \times 32$. We also empirically verify that Algorithm 1 can provide accurate solutions within a small number of iterations. We provide more details about settings and model architectures in Appendix D.

### 5.1 Classification

To check the capacity of MintNet and understand the trade-off of invertibility, we test its classification performance on MNIST and CIFAR-10, and compare it to a ResNet with a similar architecture.

On MNIST, MintNet achieves a test accuracy of 99.6%, which is the same as that of the ResNet. On CIFAR-10, MintNet reaches 91.2% test accuracy while ResNet reaches 92.6%. Both MintNet and ResNet achieve 100% training accuracy on MNIST and CIFAR-10 datasets. This indicates that MintNet has enough capacity to fit all data labels on the training dataset, and the invertible representations learned by MintNet are comparable to representations learned by non-invertible networks in terms of generalizability. Note that the small degradation in classification accuracy is also observed in other invertible networks. For example, depending on the Lipschitz constant, the gap between test accuracies of i-ResNet and RexNet can be as large as 1.92% on CIFAR-10.

Table 1: MNIST, CIFAR-10, ImageNet 32×32 bits per dimension (bpd) results. Smaller values are better. †Result not directly comparable because ZCA preprocssing was used.

| Method | MNIST | CIFAR-10 | ImageNet 32×32 |
|---|---|---|---|
| NICE [5] | 4.36 | 4.48† | - |
| MAF [24] | 1.89 | 4.31 | - |
| Real NVP [6] | 1.06 | 3.49 | 4.28 |
| Glow [16] | 1.05 | 3.35 | 4.09 |
| FFJORD [9] | 0.99 | 3.40 | - |
| i-ResNet [1] | 1.06 | 3.45 | - |
| MintNet (ours) | **0.98** | **3.32** | **4.06** |

## 5.2 Density estimation and verification of invertibility

In this section, we demonstrate the superior performance of MintNet on density estimation by training it as a flow generative model. In addition, we empirically verify that Algorithm 1 can accurately produce the inverse using a small number of iterations. We show that samples can be efficiently generated from MintNet by inverting each Mint layer with Algorithm 1.

**Density estimation.** In Tab. 1, we report bits per dimension (bpd) on MNIST, CIFAR-10, and ImageNet 32×32 datasets. It is notable that MintNet sets the new records of bpd on all three datasets. Moreover, when compared to previous best models, our MNIST model uses 30% fewer parameters than FFJORD, and our CIFAR-10 and ImageNet 32×32 models respectively use 60% and 74% fewer parameters than Glow. When trained on datasets such as CIFAR-10, MintNet requires 2 GPUs for approximately five days, while FFJORD is trained on 6 GPUs for five days, and Glow on 8 GPUs for seven days. Note that all values in Tab. 1 are with respect to the continuous distribution of uniformly dequantized images, and results of models that view images as discrete distributions are not directly comparable (*e.g.*, PixelCNN [22], IAF-VAE [17], and Flow++ [12]). To show that MintNet learns semantically meaningful representations of images, we also perform latent space interpolation similar to the interpolation experiments in Real NVP (see Appendix C).

**Verification of invertibility.** We first examine the performance of Algorithm 1 by measuring the reconstruction error of MintNets. We compute the inverse of MintNet by sequentially inverting each Mint layer with Algorithm 1. We used grid search to select the step size $\alpha$ in Algorithm 1 and chose $\alpha = 3.5, 1.1, 1.15$ respectively for MNIST, CIFAR-10 and ImageNet 32×32. An interesting fact is for MNIST, $\alpha = 3.5$ actually works better than other values of $\alpha$ within $(0, 2)$, even though it does not have the theoretical gurantee of local convergence. As Fig. 4a shows, the normalized $L_2$ reconstruction error converges within 120 iterations for all datasets considered. Additionally, Fig. 4b demonstrates that the reconstructed images look visually indistinguishable to true images.

**Samples.** Using Algorithm 1, we can generate samples efficiently by computing the inverse of MintNets. We use the same step sizes as in the reconstruction error analysis, and run Algorithm 1 for 120 iterations for all three datasets. We provide uncurated samples in Fig. 3, and more samples can be found in Appendix F. In addition, we compare our sampling time to that of the other models (see Tab. 6 in Appendix E). Our sampling method has comparable speed as i-ResNet. It is approximately 5 times faster than autoregressive sampling on MNIST, and is roughly 25 times faster on CIFAR-10 and ImageNet 32×32.

## 6 Conclusion

We propose a new method to compositionally construct invertible modules that are flexible, efficient to invert, and with a tractable Jacobian. Starting from linear transformations with triangular matrices, we apply a set of composition rules to recursively build new modules that are non-linear and more expressive (Proposition 1). We then show that the composed modules are invertible as long as their Jacobians are non-singular (Theorem 1), and propose an efficiently parallelizable numerical method

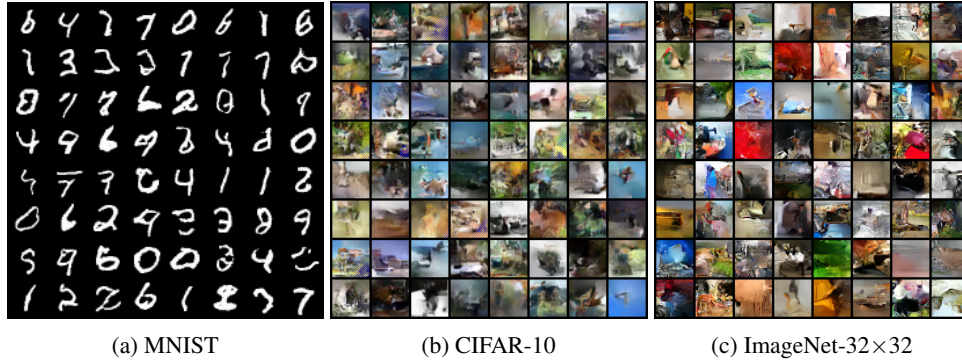

| (a) MNIST | (b) CIFAR-10 | (c) ImageNet-32×32 |

Figure 3: Uncurated samples on MNIST, CIFAR-10, and ImageNet 32×32 datasets.

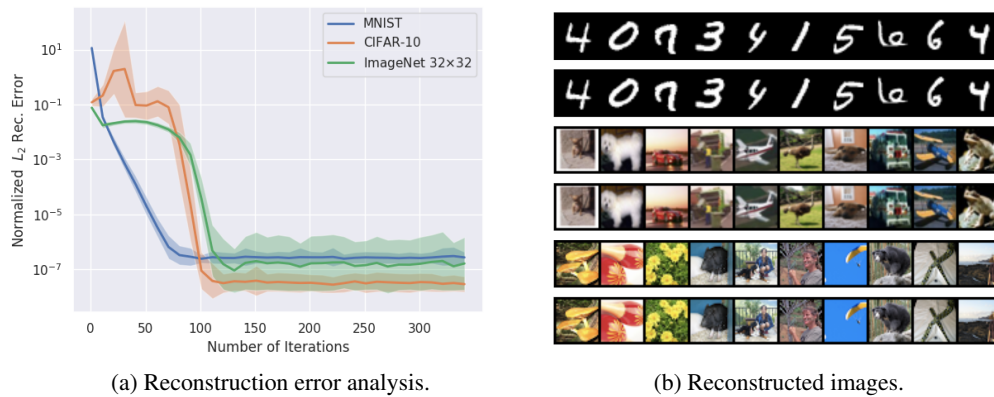

| (a) Reconstruction error analysis. | (b) Reconstructed images. |

Figure 4: Accuracy analysis of Algorithm 1 on MNIST, CIFAR-10, and ImageNet 32×32 datasets. Each curve in (a) represents the mean value of normalized reconstruction errors for 128 images. The 2nd, 4th and 6th rows in (b) are reconstructions, while other rows are original images.

(Algorithm 1) with theoretical guarantees (Theorem 2) to compute the inverse. The Jacobians of our modules are all triangular, which allows efficient and exact determinant computation.

As an application of this idea, we use masked convolutions as our basic module. Using our composition rules, we compose multiple masked convolutions together to form a module named Mint layer, following the architecture of a ResNet block. To enforce its invertibility, we constrain the masked convolutions to satisfy the condition of Theorem 1. We show that multiple Mint layers can be stacked together to form a deep invertible network which we call MintNet. The architecture can be efficiently inverted using a fixed point iteration algorithm (Algorithm 1). Experimentally, we show that MintNet performs well on MNIST and CIFAR-10 classification. Moreover, when trained as a generative model, MintNet achieves new state-of-the-art performance on MNIST, CIFAR-10 and ImageNet 32×32.

### Acknowledgements

This research was supported by Intel Corporation, Amazon AWS, TRI, NSF (#1651565, #1522054, #1733686), ONR (N00014-19-1-2145), AFOSR (FA9550- 19-1-0024).

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
