[Supplementary Material · full.pdf]

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

# A  Proofs

**Notations.** Let $J_f(\mathbf{x})$ denote the Jacobian of $f$ evaluated at $\mathbf{x}$. We use $[f(\mathbf{x})]_i$ to denote the $i$-th component of the vector-valued function $f$, and $[J_f(\mathbf{x})]_{ij}$ to denote the $ij$-th entry of $J_f(\mathbf{x})$. We further use $\mathbf{x}_i$ to denote the $i$-th component of the input vector $\mathbf{x} \in \mathbb{R}^D$, and $\frac{\partial [f(\mathbf{x})]_i}{\partial \mathbf{x}_j}\big|_{\mathbf{x}=\mathbf{t}}$ to denote the partial derivative of $[f(\mathbf{x})]_i$ w.r.t. $\mathbf{x}_j$, evaluated at $\mathbf{x} = \mathbf{t}$.

**Proposition 1.** *Define $\mathcal{F}$ as the set of all continuously differentiable functions whose Jacobian is lower triangular. Then $\mathcal{F}$ contains the basic module in Section 3.1, and is closed under the following composition rules.*

- *__Rule of addition.__ $f_1 \in \mathcal{F} \wedge f_2 \in \mathcal{F} \Rightarrow \lambda f_1 + \mu f_2 \in \mathcal{F}$, where $\lambda, \mu \in \mathbb{R}$.*

- *__Rule of composition.__ $f_1 \in \mathcal{F} \wedge f_2 \in \mathcal{F} \Rightarrow f_2 \circ f_1 \in \mathcal{F}$. A special case is $f \in \mathcal{F} \Rightarrow h \circ f \in \mathcal{F}$, where $h(\cdot)$ is a continuously differentiable non-linear activation function that is applied element-wisely.*

*Proof.* Since the basic modules have the form $f(\mathbf{x}) = \mathbf{W}\mathbf{x} + \mathbf{b}$, where $\mathbf{W}$ is a lower triangular matrix, we immediately know that $f$ is continuously differentiable and $J_f$ is lower triangular, therefore $f \in \mathcal{F}$. Next, we prove the closeness properties of $\mathcal{F}$ one by one.

- **Rule of addition**. $f = \lambda f_1 + \mu f_2$ is continuously differentiable, and $J_f$ is lower triangular. This is because $\partial f / \partial \mathbf{x} = \partial (\lambda f_1 + \mu f_2) / \partial \mathbf{x} = \lambda \partial f_1 / \partial \mathbf{x} + \mu \partial f_2 / \partial \mathbf{x}$, and both $\partial f_1 / \partial \mathbf{x}$ and $\partial f_2 / \partial \mathbf{x}$ are continuous and lower triangular.

- **Rule of composition**. $f = f_2 \circ f_1$ is continuously differentiable and has a lower triangular Jacobian. This is because $\partial f / \partial \mathbf{x} = \partial (f_2 \circ f_1) / \partial \mathbf{x} = \partial f_2 / \partial \mathbf{x}\big|_{\mathbf{x}=f_1(\mathbf{x})} \partial f_1 / \partial \mathbf{x}$, and both $\partial f_2 / \partial \mathbf{x}$ and $\partial f_1 / \partial \mathbf{x}$ are continuous and lower triangular. As a special case, we choose $f_1 = h$, where $h$ is a continuously differentiable univariate function. Since the Jacobian of $h$ is diagonal and continuous, we have $h \in \mathcal{F}$. Therefore $h \circ f_2 \in \mathcal{F}$ holds true for all $f_2 \in \mathcal{F}$.

$\square$

The following two lemmas will be very helpful for proving Theorem 1.

**Lemma 1.** *$J_f(\mathbf{x})$ is lower triangular for all $\mathbf{x} \in \mathbb{R}^D$ implies $[f(\mathbf{x})]_i$ is a function of $\mathbf{x}_1, ..., \mathbf{x}_i$, and does not depend on $\mathbf{x}_{i+1}, \cdots, \mathbf{x}_D$.*

*Proof.* Due to the fact that $J_f(\mathbf{x})$ is lower triangular, we have $[J_f(\mathbf{x})]_{i,j} = \frac{\partial [f(\mathbf{x})]_i}{\partial \mathbf{x}_j} = 0$ for any $j > i$. When $\mathbf{x}_1, ..., \mathbf{x}_{j-1}, \mathbf{x}_{j+1}, ..., \mathbf{x}_D$ are fixed, we have

$$[f(\mathbf{x}_1, ..., \mathbf{x}_{j-1}, \mathbf{x}_j, \mathbf{x}_{j+1}, \mathbf{x}_D)]_i = [f(\mathbf{x}_1, ..., \mathbf{x}_{j-1}, 0, \mathbf{x}_{j+1}, ..., \mathbf{x}_D)]_i + \int_0^{\mathbf{x}_j} \frac{\partial [f(\mathbf{t})]_i}{\partial \mathbf{t}_j} \mathrm{d}\mathbf{t}_j \quad (5)$$

$$= [f(\mathbf{x}_1, ..., \mathbf{x}_{j-1}, 0, \mathbf{x}_{j+1}, ..., \mathbf{x}_D)]_i. \quad (6)$$

This implies that $[f(\mathbf{x})]_i$ does not depend on $\mathbf{x}_j$ for any $j > i$. In other words, $f(\mathbf{x})$ is only a function of $\mathbf{x}_1, ..., \mathbf{x}_i$. $\square$

**Lemma 2.** $\mathrm{diag}(J_f(\mathbf{x}) J_f(\mathbf{0})) > \mathbf{0}$ *implies that for any $1 \le i \le n$, either (i) $\forall \mathbf{x} \in \mathbb{R}^D : [J_f(\mathbf{x})]_{ii} > 0$ or (ii) $\forall \mathbf{x} \in \mathbb{R}^D : [J_f(\mathbf{x})]_{ii} < 0$. That is, $[f(\mathbf{x})]_i$ is monotonic w.r.t. $\mathbf{x}_i$ when $\mathbf{x}_1, \cdots, \mathbf{x}_{i-1}$ are fixed.*

*Proof.* Clearly $\mathrm{diag}(J_f(\mathbf{x}) J_f(\mathbf{0})) > \mathbf{0}$ is equivalent to $\forall 1 \le i \le n, \mathbf{x} \in \mathbb{R}^D : [J_f(\mathbf{x})]_{ii} [J_f(\mathbf{0})]_{ii} = \frac{\partial [f(\mathbf{x})]_i}{\partial \mathbf{x}_i} \frac{\partial [f(\mathbf{x})]_i}{\partial \mathbf{x}_i}\big|_{\mathbf{x}=\mathbf{0}} > 0$. This means for any $\mathbf{x} \in \mathbb{R}^D$, $[J_f(\mathbf{x})]_{ii} = \frac{\partial [f(\mathbf{x})]_i}{\partial \mathbf{x}_i} \ne 0$ and it shares the same sign with $[J_f(\mathbf{0})]_{ii} = \frac{\partial [f(\mathbf{x})]_i}{\partial \mathbf{x}_i}\big|_{\mathbf{x}=\mathbf{0}}$, a constant that is either strictly positive or strictly negative. This further implies that when $\mathbf{x}_1, \cdots, \mathbf{x}_{i-1}$ are fixed, $\frac{\partial [f(\mathbf{x})]_i}{\partial \mathbf{x}_i}$ is either strictly positive or strictly negative for all $\mathbf{x}_i \in \mathbb{R}$, and $[f(\mathbf{x})]_i$ is therefore monotonic w.r.t. $\mathbf{x}_i$. $\square$

**Theorem 1.** *If $f \in \mathcal{F}$ and $J_f(\mathbf{x})$ is non-singular for all $\mathbf{x}$ in the domain, then $f$ is invertible.*

*Proof.* Assume without loss of generality that $J_f(\mathbf{x})$ is lower triangular. We first prove that $\text{diag}(J_f(\mathbf{x})J_f(\mathbf{0})) > \mathbf{0}$ by contradiction. Assuming $\text{diag}(J_f(\mathbf{x})J_f(\mathbf{0})) \leq \mathbf{0}$, then $\exists 1 \leq i \leq n, \mathbf{x}' \in \mathbb{R}^D$ such that $[J_f(\mathbf{x}')]_{ii}[J_f(\mathbf{0})]_{ii} \leq 0$. Because $J_f(\mathbf{x})$ is always triangular and non-singular, we immediately conclude that $[J_f(\mathbf{x}')]_{ii}[J_f(\mathbf{0})]_{ii} < 0$. Assume without loss of generality that $[J_f(\mathbf{0})]_{ii} > 0$ and $[J_f(\mathbf{x}')]_{ii} < 0$. Then, by the intermediate value theorem, we know that $\exists t \in (0, 1)$ such that $[J_f(t\mathbf{x}')]_{ii} = 0$, which contradicts that fact that $J_f(\mathbf{x})$ is always non-singular.

Next, we prove that for all $\mathbf{z}$ in the range of $f(\mathbf{x})$, there exists a unique $\mathbf{x}$ such that $f(\mathbf{x}) = \mathbf{z}$. To obtain $\mathbf{x}_1$, we only need to solve $[f(\mathbf{x})]_1 = \mathbf{z}_1$, which is an equation of variable $\mathbf{x}_1$, as concluded from Lemma 1. Since Lemma 2 implies that $[f(\mathbf{x})]_1$ is monotonic w.r.t. $\mathbf{x}_1$, we know that $[f(\mathbf{x})]_1$ has a unique inverse $\mathbf{x}_1$ whenever $\mathbf{z}_1$ is in the range of $[f(\mathbf{x})]_1$. Now assume we have already obtained $\mathbf{x}_1, ..., \mathbf{x}_k$, where $k \geq 1$. In this case, Lemma 1 asserts that $[f(\mathbf{x})]_{k+1} = \mathbf{z}_{k+1}$ is an equation of variable $\mathbf{x}_{k+1}$. Again Lemma 2 implies that $[f(\mathbf{x})]_{k+1}$ is a monotonic function of $\mathbf{x}_{k+1}$ given $\mathbf{x}_1, \cdots, \mathbf{x}_k$, which implies further that $[f(\mathbf{x})]_{k+1} = \mathbf{z}_{k+1}$ has a unique solution $\mathbf{x}_{k+1}$ whenever $\mathbf{z}_{k+1}$ is in the range of $[f(\mathbf{x})]_{k+1}$. By induction, we can solve for $\mathbf{x}_1, \mathbf{x}_2, \cdots, \mathbf{x}_D$ by repetitively employing this procedure, which concludes that $f^{-1}(\mathbf{z}) = (\mathbf{x}_1, ..., \mathbf{x}_D)^\mathsf{T}$ exists, and can be determined uniquely.

$\square$

**Theorem 2.** *The iterative method of Algorithm 1 is locally convergent whenever $0 < \alpha < 2$.*

*Proof.* Let $\mathbf{z}$ be any value in the range of $f(\mathbf{x})$ and $g(\mathbf{x}; \alpha, \mathbf{z}) \triangleq \mathbf{x} - \alpha \, \text{diag}(J_f(\mathbf{x}))^{-1}[f(\mathbf{x}) - \mathbf{z}]$, where $\text{diag}(A)^{-1}$ denotes a diagonal matrix whose diagonal entries are the reciprocals of those of $A$. The iterative method of Algorithm 1 can be written as $\mathbf{x}_t = g(\mathbf{x}_{t-1}; \alpha, \mathbf{z})$. Because of Theorem 1, there exists a unique $\mathbf{x}^* \in \mathbb{R}^D$ such that $f(\mathbf{x}^*) = \mathbf{z}$, in which case $g(\mathbf{x}^*; \alpha, \mathbf{z}) = \mathbf{x}^*$. Applying the product rule, we have

$$J_g(\mathbf{x}^*; \alpha, \mathbf{z}) = I - \alpha \, \text{diag}(J_f(\mathbf{x}^*))^{-1} J_f(\mathbf{x}^*),$$

where $J_g(\mathbf{x}^*; \alpha, \mathbf{z})$ denotes the Jacobian of $g(\mathbf{x}; \alpha, \mathbf{z})$ evaluated at $\mathbf{x}^*$. Since $J_f(\mathbf{x}^*)$ is triangular, $J_g(\mathbf{x}^*; \alpha, \mathbf{z})$ will also be triangular. Therefore, the only eigenvalue of $J_g(\mathbf{x}^*; \alpha, \mathbf{z})$ is $1 - \alpha$, due to the fact that the only solution to the equation system $\det(\lambda I - J_g(\mathbf{x}^*; \alpha, \mathbf{z})) = (\lambda - 1 + \alpha)^D = 0$ is $\lambda = 1 - \alpha$. Since $0 < \alpha < 2$, the spectral radius of $J_g(\mathbf{x}^*; \alpha, \mathbf{z})$ satisfies $\rho(J_g(\mathbf{x}^*; \alpha, \mathbf{z})) = |1 - \alpha| < 1$. Then the Ostrowski Theorem (*cf.*, Theorem 10.1.3. in [23]) shows that the sequence $\{\mathbf{x}_1, \mathbf{x}_2, \cdots, \mathbf{x}_t\}$ obtained by $\mathbf{x}_t = g(\mathbf{x}_{t-1}; \alpha, \mathbf{z})$ converges locally to $\mathbf{x}^*$ as $t \to \infty$. $\square$

## B  Masked convolutions

Convolution is a special type of linear transformation that proves to be very effective for image data. The basic invertible module can be implemented using masked convolutions (*e.g.*, causal convolutions in PixelCNN [22]). Consider a 2D convolutional layer with $C_{\text{in}}$ input feature maps, $C_{\text{out}}$ filters, a kernel size of $R \times R$ and a zero-padding of $\lfloor R/2 \rfloor$. We assume $R$ is an odd integer and $C_{\text{out}} = C_{\text{in}}$ so that the input and output of the convolutional layer have the same shape. Let $\mathbf{W} \in \mathbb{R}^{C_{\text{out}} \times C_{\text{in}} \times R \times R}$ be the weight tensor of this layer. We define a mask $\mathbf{M} \in \{0, 1\}^{C_{\text{out}} \times C_{\text{in}} \times R \times R}$ that satisfies

$$\mathbf{M}[i, j, m, n] = \begin{cases} 0, & \text{if } i < j \text{ or } i = j \wedge m > \lfloor R/2 \rfloor \text{ or } i = j \wedge m = \lfloor R/2 \rfloor \wedge n > \lfloor R/2 \rfloor, \\ 1, & \text{Otherwise.} \end{cases} \quad (7)$$

The masked convolution then uses $\mathbf{M} \odot \mathbf{W}$ as the weight tensor. In Fig. 1, we provide an illustration on a $3 \times 3$ masked convolution with 3 filters.

In MintNet, $\mathfrak{L}(\mathbf{x})$ is efficiently implemented with 3 masked convolutional layers. The weights and masks are denoted as $(\mathbf{W}^1, \mathbf{M}^1)$, $(\mathbf{W}^2, \mathbf{M}^2)$ and $(\mathbf{W}^3, \mathbf{M}^3)$, which separately correspond to $\{\mathbf{W}_i^1\}_{i=1}^K$, $\{\mathbf{W}_{ij}^2\}_{1 \leq i,j \leq K}$, $\{\mathbf{W}_j^3\}_{j=1}^K$ in Eq. (2). Let $C$ be the number of input feature maps, and suppose the kernel size is $R \times R$. The shapes of $\mathbf{W}^1$, $\mathbf{W}^2$ and $\mathbf{W}^3$ are respectively $(KC, C, R, R)$, $(KC, KC, R, R)$ and $(C, KC, R, R)$. The masks of them are simple concatenations of copies of the mask in Eq. (7). For instance, $\mathbf{M}^1$ consists of $K$ copies of Eq. (7), and $\mathbf{M}^2$ consists of $K \times K$ copies.

Figure 5: MintNet interpolation of hidden representation. **Left:** MintNet MNIST latent space interpolation. **Middle:** MintNet CIFAR-10 latent space interpolation. **Right:** MintNet ImageNet 32×32 latent space interpolation.

Using masked convolutions, $\mathfrak{L}(\mathbf{x})$ can be concisely written as

$$\mathfrak{L}(\mathbf{x}) = \mathbf{t} \odot \mathbf{x} + (\mathbf{W}^3 \odot \mathbf{M}^3) \circledast h\bigg( (\mathbf{W}^2 \odot \mathbf{M}^2) \circledast h\big((\mathbf{W}^1 \odot \mathbf{M}^1) \circledast \mathbf{x} + \mathbf{b}^1\big) + \mathbf{b}^2 \bigg) + \mathbf{b}^3, \quad (8)$$

where $\mathbf{b}^1, \mathbf{b}^2, \mathbf{b}^3$ are biases, and $\circledast$ denotes the operation of discrete 2D convolution.

## C   Interpolation of hidden representations

Given four images $\mathbf{x}_1, \mathbf{x}_2, \mathbf{x}_3, \mathbf{x}_4$ in the dataset, let $\mathbf{z}_i = f(\mathbf{x}_i)$, where $i = 1, 2, 3, 4$, be the corresponding features in the feature domain. Similar to [6], in the feature domain, we define

$$\mathbf{z} = \cos(\phi)(\cos(\phi')\mathbf{z}_1 + \sin(\phi')\mathbf{z}_2) + \sin(\phi)(\cos(\phi')\mathbf{z}_3 + \sin(\phi')\mathbf{z}_4) \quad (9)$$

where $\mathbf{x}$-axis corresponds to $\phi'$, $\mathbf{y}$-axis corresponds to $\phi$, and both $\phi$ and $\phi'$ range over $\{0, \frac{\pi}{14}, ..., \frac{7\pi}{14}\}$. We then transform $\mathbf{z}$ back to the image domain by taking $f^{-1}(\mathbf{z})$. Interpolation results are shown in Fig. 5.

## D   Experiment setup and network architecture

**Hyperparameter tuning and computation infrastructure.**    We use the standard train/test split of MNIST, CelebA and CIFAR-10. We tune our models by observing its training bpd. For density estimation on CIFAR-10 and ImageNet 32×32, the models were run on two Titan XP GPUs. In other cases the model was run on one Titan XP GPU.

**Classification setup.**    Following [1], we pad the images to 16 channels with zeros. This corresponds to the first convolution in ResNet which increases the number of channels to 16. Both ResNet and our MintNet are trained with AMSGrad [26] for 200 epochs with the cosine learning rate schedule [19] and an initial learning rate of 0.001. Both networks use a batch size of 128.

**Classification architecture.**    The ResNet contains 38 pre-activation residual blocks [11], and each block has three $3 \times 3$ convolutions. The architecture is divided into 3 stages, with 16, 64 and 256 filters respectively. Our MintNet uses 19 grouped invertible layers, which include a total of 38 residual invertible layers, each having three $3 \times 3$ convolutions. Batch normalization is applied before each invertible layer. Note that batch normalization does not affect the invertibility of our network, because during test time it uses fixed running average and standard deviation and is an invertible operation. We use 2 squeezing blocks at the same position where ResNet applies subsampling, and matches the number of filters used in ResNet. To produce the logits for classification, both MintNet and ResNet first apply global average pooling and then use a fully connected layer (see Tab. 2).

**Density estimation setup.**    We mostly follow the settings in [24]. All training images are dequantized and transformed using the logit transformation. Networks are trained using AMSGrad [25].

On MNIST, we decay the learning rate by a factor of 10 at the 250th and 350th epoch, and train for 400 epochs. On CIFAR-10, we train with cosine learning rate decay for a total of 200 epochs. On ImageNet 32×32, we train with cosine learning rate decay for a total of 350k steps. All initial learning rates are 0.001.

**Density estimation architecture.** For density estimation on MNIST, we use 20 paired Mint layers with 45 filters each. For both CIFAR-10 and ImageNet 32×32, we use 21 paired Mint layers, each of which has 255 filters. For all the three datasets, two squeezing operations are used and are distributed evenly across the network (see Tab. 3 and Tab. 4).

**Tuning the step size for sampling.** We perform grid search to find hyperparamter $\alpha$ for Algorithm 1 using a minibatch of 128 images. More specifically, we start from $\alpha = 1$ to 5 with a step size 0.5 for MNIST, CIFAR-10, and ImageNet 32×32, and compute the normalized $L_2$ reconstruction error with respect to the number of iterations. The normalized $L_2$ error is defined as $\|\mathbf{x} - \mathbf{y}\|_2^2 / D$, where $\mathbf{x} \in \mathbb{R}^D$ and $\mathbf{y} \in \mathbb{R}^D$ are two image vectors corresponding to the original and reconstructed images. We find that the algorithm converges most quickly when $\alpha$ is in intervals $[3, 4]$, $[1, 2]$ and $[1, 2]$ for MNIST, CIFAR-10 and ImageNet 32×32 respectively. Then we perform a second round grid search on the corresponding interval with a step size 0.05. In this case, we are able to find the best $\alpha$, that is $\alpha = 3.5, 1.1, 1.15$ for the corresponding datasets.

**Verification of invertibility.** To verify the invertibility of MintNet, we study the normalized $L_2$ reconstruction error for MNIST, CIFAR-10 and ImageNet 32×32. The $L_2$ reconstruction error is computed for 128 images on all three datasets. We plot the exponential of the mean log reconstruction errors in Fig. 4. The shaded area corresponds to the exponential of the standard deviation of log reconstruction errors.

Table 2: MintNet image classification network architecture.

| Name | Configuration | Replicate Block |
|------|---------------|-----------------|
| Paired Mint Block1 with Batch Normalization | batch normalization<br>$3 \times 3$ lower triangular masked convolution, 1 filter<br>leaky relu activation<br>$3 \times 3$ lower triangular masked convolution, 1 filter<br>leaky relu activation<br>$3 \times 3$ lower triangular masked convolution, 1 filter<br>batch normalization<br>$3 \times 3$ upper triangular masked convolution,1 filter<br>leaky relu activation<br>$3 \times 3$ upper triangular masked convolution, 1 filter<br>leaky relu activation<br>$3 \times 3$ upper triangular masked convolution, 1 filter | $\times 6$ |
| Squeezing Layer | $2 \times 2$ squeezing layer | — |
| Paired Mint Block2 with Batch Normalization | batch normalization<br>$3 \times 3$ lower triangular masked convolution, 1 filter<br>leaky relu activation<br>$3 \times 3$ lower triangular masked convolution, 1 filter<br>leaky relu activation<br>$3 \times 3$ lower triangular masked convolution, 1 filter<br>batch normalization<br>$3 \times 3$ upper triangular masked convolution,1 filter<br>leaky relu activation<br>$3 \times 3$ upper triangular masked convolution, 1 filter<br>leaky relu activation<br>$3 \times 3$ upper triangular masked convolution, 1 filter | $\times 6$ |
| Squeezing Layer | $2 \times 2$ squeezing layer | — |
| Paired Mint Block3 with Batch Normalization | batch normalization<br>$3 \times 3$ lower triangular masked convolution, 1 filter<br>leaky relu activation<br>$3 \times 3$ lower triangular masked convolution, 1 filter<br>leaky relu activation<br>$3 \times 3$ lower triangular masked convolution, 1 filter<br>batch normalization<br>$3 \times 3$ upper triangular masked convolution,1 filter<br>leaky relu activation<br>$3 \times 3$ upper triangular masked convolution, 1 filter<br>leaky relu activation<br>$3 \times 3$ upper triangular masked convolution, 1 filter | $\times 7$ |
| Output Layer | average pooling<br>fully connected layer<br>softmax layer | — |

Table 3: MintNet MNIST density estimation network architecture.

| Name | Configuration | Replicate Block |
|------|:-------------:|:---------------:|
| Paired Mint Block1 | $3 \times 3$ lower triangular masked convolution, 45 filters<br>elu activation<br>$3 \times 3$ lower triangular masked convolution, 45 filters<br>elu activation<br>$3 \times 3$ lower triangular masked convolution, 45 filters<br>$3 \times 3$ upper triangular masked convolution, 45 filters<br>elu activation<br>$3 \times 3$ upper triangular masked convolution, 45 filters<br>elu activation<br>$3 \times 3$ upper triangular masked convolution, 45 filters | $\times 6$ |
| Squeezing Layer | $2 \times 2$ squeezing layer | — |
| Paired Mint Block2 | $3 \times 3$ lower triangular masked convolution, 45 filters<br>elu activation<br>$3 \times 3$ lower triangular masked convolution, 45 filters<br>elu activation<br>$3 \times 3$ lower triangular masked convolution, 45 filters<br>$3 \times 3$ upper triangular masked convolution, 45 filters<br>elu activation<br>$3 \times 3$ upper triangular masked convolution, 45 filters<br>elu activation<br>$3 \times 3$ upper triangular masked convolution, 45 filters | $\times 6$ |
| Squeezing Layer | $2 \times 2$ squeezing layer | — |
| Paired Mint Block3 | $3 \times 3$ lower triangular masked convolution, 45 filters<br>elu activation<br>$3 \times 3$ lower triangular masked convolution, 45 filters<br>elu activation<br>$3 \times 3$ lower triangular masked convolution, 45 filters<br>$3 \times 3$ upper triangular masked convolution, 45 filters<br>elu activation<br>$3 \times 3$ upper triangular masked convolution, 45 filters<br>elu activation<br>$3 \times 3$ upper triangular masked convolution, 45 filters | $\times 8$ |

Table 4: MintNet CIFAR-10 and Imagenet $32\times32$ density estimation network architecture.

| Name | Configuration | Replicate Block |
|---|---|---|
| Paired Mint Block1 | $3 \times 3$ lower triangular masked convolution, 85 filters<br>elu activation<br>$3 \times 3$ lower triangular masked convolution, 85 filters<br>elu activation<br>$3 \times 3$ lower triangular masked convolution, 85 filters<br>$3 \times 3$ upper triangular masked convolution,85 filters<br>elu activation<br>$3 \times 3$ upper triangular masked convolution, 85 filters<br>elu activation<br>$3 \times 3$ upper triangular masked convolution, 85 filters | $\times 7$ |
| Squeezing Layer | $2 \times 2$ squeezing layer | — |
| Paired Mint Block2 | $3 \times 3$ lower triangular masked convolution, 85 filters<br>elu activation<br>$3 \times 3$ lower triangular masked convolution, 85 filters<br>elu activation<br>$3 \times 3$ lower triangular masked convolution, 85 filters<br>$3 \times 3$ upper triangular masked convolution,85 filters<br>elu activation<br>$3 \times 3$ upper triangular masked convolution, 85 filters<br>elu activation<br>$3 \times 3$ upper triangular masked convolution, 85 filters | $\times 7$ |
| Squeezing Layer | $2 \times 2$ squeezing layer | — |
| Paired Mint Block3 | $3 \times 3$ lower triangular masked convolution, 85 filters<br>elu activation<br>$3 \times 3$ lower triangular masked convolution, 85 filters<br>elu activation<br>$3 \times 3$ lower triangular masked convolution, 85 filters<br>$3 \times 3$ upper triangular masked convolution,85 filters<br>elu activation<br>$3 \times 3$ upper triangular masked convolution, 85 filters<br>elu activation<br>$3 \times 3$ upper triangular masked convolution, 85 filters | $\times 7$ |

# E  Additional tables

Table 5: Comparison to some common invertible models.

| Property | NICE | Real-NVP | Glow | MaCow | FFJORD | i-ResNet | MintNet |
|---|---|---|---|---|---|---|---|
| Analytic Forward | ✓ | ✓ | ✓ | ✓ | ✗ | ✓ | ✓ |
| Analytic Inverse | ✓ | ✓ | ✗ | ✗ | ✗ | ✗ | ✗ |
| Non-volume Preserving | ✗ | ✓ | ✓ | ✓ | ✓ | ✓ | ✓ |
| Exact Likelihood | ✓ | ✓ | ✓ | ✓ | ✗ | ✗ | ✓ |

Table 6: Sampling time for 64 samples for MintNet, i-ResNet and autoregressive method on the same model architectures. The time is evaluated on a NVIDIA TITAN Xp.

| Method | MNIST | CIFAR-10 | ImageNet $32\times32$ |
|---|---|---|---|
| i-ResNet [1] (100 iterations) | 11.56s | 99.41s | 92.53s |
| Autoregressive (1 iteration) | 63.61s | 2889.64s | 2860.21s |
| MintNet (120 iterations) (ours) | 12.81s | 117.83s | 120.78s |

# F More Samples

In this section, we provide more uncurated MintNet samples on MNIST, CIFAR-10 and ImageNet 32×32.

Figure 6: MintNet MNIST samples.

Figure 7: MintNet CIFAR-10 samples.

Figure 8: MintNet ImageNet 32×32 samples.