[Reviews · NeurIPS 2019]

Reviewer 1



Summary: The authors suggest a new composition rule that brings flow models and autoregressive closer together. The main contribution is a sufficient condition for invertibility and a type of masked convolution design with triangular Jacobian, whose inverse can be computed sequentially in a parallelizable manner. Some concerns remain about novelty. Pros: There is a dire need for a generalized framework around recent progress in flow-based modeling. The paper aims to provide this. The empirical results are strong. Major concerns: A recent paper [1] is neither cited nor discussed. However, the paper contains masked convolutions and discusses relationship to autoregressive models. It also has to compute a sequential, but parallelizable inverse. I am concerned about the novelty of the current paper and its lack of discussion of this work. If the authors can not make a compeling argument how their method is sufficiently more general or different than [1], the current work seems to be too incremental to warrant acceptance to Neurips. [1] Hoogeboom et al., ICML 2019, Emerging Convolutions for Generative Normalizing Flows. ==== Post rebuttal ==== The authors addressed my major concern and discussed emerging convolutions in the context of their work. The significantly reduced time-complexity due to the fixed point iteration inverse is a significant improvement, so I will raise my score by one point.

Reviewer 2



The paper is fairly clear. Experimental results show promises of the proposed architecture and the module along with its inference/reversed inference seems simple enough to be practical. However, the main reasons for my borderline score are: (1) significance and novelty of the proposed method: existing works such as Flow++ is already using similar idea and achieving better results (the authors mention Flow++ treat the output is discrete, but the gap is quite significant). (2) experimental results: the qualitative results are not too impressive and ImageNet only is up to 32x32. Appendix mentioned CelebA but no results. Additionally, the math proofs in Appendix (i.e. Thm1) is classic linear algebra results. The authors can simply cite e.g. Matrix Analysis, and give a re-phrased proof in context for clarity. I raised my score to 6 on the author's promise to release code.

Reviewer 3



Originality: While I would not call the use of masked transformations particularly novel in this setting, the authors present a satisfying and simple architecture which should be broadly applicable to many domains and tasks. This stands in contrast to many other invertible models which utilize very tailored and domain specific architectures. Quality: I believe this paper to be of high quality. The strong performance of the proposed architecture on generative modeling is well-backed by experimental results. I feel the classification experiments could have been stronger and presented more clearly. The proposed architecture can be implemented with an identical structure to an iResNet (by replacing the masked convolutions with spectral-normalized convolutions). A comparison of this kind could improve the strength of the author's claims. I was also somewhat confused about the effect of the step size in the inverse algorithm. Given the authors grid-searched over this parameter, it seems important. I do not feel like the authors adequately explained the impact of this parameter on reconstruction accuracy and speed. The performance of the method presented by the authors to invert their layers relies on the choice of an initial guess of the solution x_0. The authors do not present a method for choosing this initial guess and they also do not empirically show the effect of making a bad choice. In lieu of theoretical claims about this, some further experiments would be sufficient. Clarity: Overall the paper is clear and well-written. The introduction and background sections motivate the development of invertible models and give the reader a solid background on related methods. In section 3 the authors introduce their approach and discuss how many previous methods fall into their framework. The authors introduce some compositionally rules that all one to define invertible models. These rules are clear, as is their proof. I was somewhat confused by some aspects of the chosen model architecture. How are "paired" mint layers created? Is an upper-triangular layer followed by a lower-triangular layer? Are their results added together? I found this unclear. The authors introduce paired layers to combat a potential argument about the choice of feature ordering used. Does using paired layers improve performance compared to non-paired layers? Significance: The biggest contribution of this work is the attempt to formalize and standardize the design of invertible models. Most previous work in the area has been very ad-hoc. If bijective models become a popular model paradigm then the ideas presented in this paper may be quite useful. Similar standardization ideas were presented in the iResNet paper. The biggest advantage this work has over that work is the ability to compute the jacobian log-determinant exactly. This is appealing to have but further work has demonstrated that improved performance can be obtained on generative modeling without exact log-determinant computation. -------- post rebuttal ------- I thank the authors for clarifying on a few questions I had. This is appreciated but I will maintain my score.

[Author Response · NeurIPS 2019]

We thank all the reviewers for providing constructive feedback. First, we hope to bring a new theoretical result to the
reviewers' attention. We can now prove that **Algorithm 1 is guaranteed to converge locally** whenever $0 < \alpha < 2$.
We will include this new result in the revision.

**To Reviewer #1**

**Q1:** *Difference from emerging convolutions.*

We thank R1 for pointing out a missing reference. We developed our approach without knowing the work of emerging
convolutions. We will include a discussion of it in Section 4.3. We believe that our approach has *critical advantages*
*over emerging convolutions*, which can be detailed as follows:

- 9 **The inversion is much faster to compute.** Emerging convolutions are inverted using forward/back substi-
tutions designed for inverting triangular matrices, which requires the same number of iterations as the input
dimension. In stark contrast, we use a fixed-point iteration method (Algorithm 1) for inversion, which is similar
to i-ResNet and requires substantially fewer iterations than the input dimension. For example, our inversion
takes 120 iterations to converge on CIFAR-10, while emerging convolutions will need 3072 iterations. In other
words, our inversion is roughly 25 times faster than emerging convolutions on powerful GPUs.

- 15 **No bottleneck in the architecture.** The input and output of an emerging convolution must have the same
dimensions, and therefore emerging convolutions cannot be used to expand the hidden dimensions of a neural
network. As a result, if a flow model is built using only emerging convolutions, the largest number of hidden
units for a layer will be the input dimension, which becomes a bottleneck in expressive power. In stark contrast,
we can use masked convolutions to expand the hidden dimensions (See Eq. (2)) and eliminate the bottleneck.
This is why we can build MintNet *using only masked convolutions*, while the emerging convolution paper
modified the Glow architecture by replacing $1 \times 1$ convolutions with emerging convolutions.

**Q2:** *Strengths over i-ResNet, ResNet, Flow++, etc.*

We would like to remind the reviewer that we use Section 4.3 to compare MintNet with many popular paradigms of flow
models, *e.g.*, free-form invertible models (including i-ResNet) and dimension partitioning models (including Flow++).
We will clarify Section 4.3 more to incorporate the following information.

Unlike ResNet, MintNet is an invertible architecture that can be used directly as a flow generative model for density
estimation and sample generation. Unlike i-ResNet, MintNet provides exact likelihood values and better empirical
performance (see Table 1), while having a comparable cost of sample generation. Unlike Flow++, MintNet is a
free-form invertible model that does not rely on dimension partitioning and coupling layers. We did not compare results
with Flow++, as it uses variational dequantization, while all models we compare (Glow, i-ResNet, and other models in
Table 1) use uniform dequantization. For the same model, the bpd values resulting from variational dequantization are
**lower bounds** to the bpd values of uniform dequantization, and therefore will always appear to be "better" (hence not
comparable).

**To Reviewer #2**

**Q1:** *Code and experiments on ImageNet* $64 \times 64$.

We will put the code and checkpoints online once the paper is accepted. We agree that results on more datasets are
helpful. However, our resources are too limited to finish experiments on ImageNet $64 \times 64$, and we believe that our
state-of-the-art results across all three datasets (MNIST, CIFAR-10, and ImageNet $32 \times 32$) are sufficient to demonstrate
the advantages of MintNet.

**To Reviewer #3**

**Q1:** *The initial choice of* $\mathbf{x}_0$.

When inverting Mint layers, we always choose $\mathbf{x}_0 = \mathbf{z} \odot \frac{1}{\mathbf{t}}$ because we empirically observe $\mathbf{z} = \mathfrak{L}(\mathbf{x}) \approx \mathbf{t} \odot \mathbf{x}$ for
deeper layers after training. We will elaborate more on this in Section 4.1.

**Q2:** *More explanations on Paired Mint layers.*

A Paired Mint layer is a lower triangular Mint layer followed by an upper triangular Mint layer. The results of the lower
triangular layer are fed to the upper one as the input. There are clearer descriptions on Paired Mint layers in the network
architecture tables in Appendix D (see, *e.g.*, Table 3). We will clarify this more in Section 4.2.

[Meta-Review · NeurIPS 2019]

This paper received OK scores overall, with little disparity in the final scores. There is consensus among reviewers that this paper is well-written, clear and that the experiments are well-designed. Novelty is a weak point of this work, which is more of a framework paper.